# PRACTICAL ADVERSARIAL ATTACKS ON STOCHASTIC BANDITS VIA FAKE DATA INJECTION

## ABSTRACT

Adversarial attacks on stochastic bandits have traditionally relied on some unrealistic assumptions, such as per-round reward manipulation and unbounded perturbations, limiting their relevance to real-world systems. We propose a more practical threat model, Fake Data Injection, which reflects realistic adversarial constraints: the attacker can inject only a limited number of bounded fake feedback samples into the learner's history, simulating legitimate interactions. We design efficient attack strategies under this model, explicitly addressing both magnitude constraints (on reward values) and temporal constraints (on when and how often data can be injected). Our theoretical analysis shows that these attacks can mislead both Upper Confidence Bound (UCB) and Thompson Sampling algorithms into selecting a target arm in nearly all rounds while incurring only sublinear attack cost. Experiments on synthetic and real-world datasets validate the effectiveness of our strategies, revealing significant vulnerabilities in widely used stochastic bandit algorithms under practical adversarial scenarios.

## 1 INTRODUCTION

Multi-armed bandit (MAB) algorithms are widely used in online decision-making systems for their ability to balance exploration and exploitation using partial feedback. They form the backbone of many interactive applications, including personalized recommendation Li et al. (2010), online advertising Chen et al. (2016), clinical trials Villar et al. (2015), and adaptive routing Li et al. (2016). As these algorithms are increasingly deployed in high-stakes, user-facing systems, growing concerns have emerged regarding their vulnerability to adversarial manipulation. A growing body of work Jun et al. (2018); Liu and Shroff (2019); Zuo et al. (2023); Zuo (2024) has shown that MAB algorithms are vulnerable to adversarial attacks on feedback, where an attacker subtly perturbs the observed rewards to mislead the learning process. Remarkably, even with limited intervention, the attacker can steer the learner toward selecting a targeted but suboptimal arm in the vast majority of rounds.

However, prior works on adversarial attacks against stochastic bandits (Jun et al., 2018; Liu and Shroff, 2019; Zuo, 2024) almost exclusively adopt a *feedback-perturbation* threat model, where the attacker observes the learner's chosen arm and the corresponding reward in each round, then arbitrarily modifies that reward before it is revealed to the learner. This assumption effectively grants the attacker *full control of the feedback channel at every timestep*—a level of adversarial power that is unrealistic in real-world systems. For example, in a restaurant recommendation platform, no adversary can continuously overwrite genuine user ratings submitted by thousands of real customers. At most, they can register fake accounts and inject fabricated reviews. Likewise, in online advertising or click-through prediction, practical attacks take the form of click fraud or synthetic interactions, which add fake feedback rather than altering genuine user data. In other words, prior models *overestimate* the attacker's per-round control, while simultaneously *underestimating* their ability to inject feedback across arbitrary arms. These mismatches motivate our shift to a more realistic and constrained threat model: the *Fake Data Injection* model.

Building on these observations, we propose a more realistic and practically grounded threat model: the *Fake Data Injection* model. Instead of modifying genuine feedback, the attacker influences the learner indirectly by injecting a limited number of fabricated (arm, reward) pairs into its interaction history. These fake samples must conform to valid feedback ranges (e.g., binary clicks or 1–5 star ratings), and their injection is subject to constraints such as system-level detection or resource limits.

The learner processes these fake interactions indistinguishably from real ones—updating estimates, counts, and decision logic accordingly.

This model captures practical attack surfaces overlooked by previous works. It removes the unrealistic assumption of per-round reward manipulation, enables feedback injection on arbitrary arms, and respects the bounded nature of real-world feedback. At the same time, it introduces new algorithmic challenges that cannot be addressed by existing techniques. Unlike the standard model—where the attacker perturbs rewards in real time—fake data injection raises fundamental questions about *when*, *how strongly*, and *how frequently* to inject samples to effectively influence the learner. The attacker must decide: (i) how many fake samples are required to suppress the selections of a non-target arm, (ii) how to achieve this using only bounded reward values, and (iii) how to distribute injections over time when batch size or injection frequency is constrained. These challenges call for new analytical tools and attack strategies that explicitly account for both *magnitude* constraints (on reward values) and *temporal* constraints (on when and how often data can be injected).

**Our Contributions.** We develop a suite of attack strategies tailored to the fake data injection model, addressing both theoretical and practical challenges:

- We propose the *Least Injection* algorithm for the *unbounded* setting, showing that a single fake sample per non-target arm suffices to steer the learner toward a target arm with sublinear cost. A key technical tool is the *Exponential Suppression Lemma*, which ensures long-term suppression of non-target arms and guides the design of our subsequent algorithms.

- We extend this to the *bounded* setting via the *Simultaneous Bounded Injection (SBI)* algorithm, which replicates the effect of an unbounded sample using a batch of bounded fake data, while maintaining sublinear cost.

- To address stricter constraints, we propose the *Periodic Bounded Injection (PBI)* algorithm, which injects small batches at controlled intervals. We provide a new suppression analysis to guarantee its effectiveness under temporal and magnitude constraints.

- All attack algorithms are analyzed under both UCB and Thompson Sampling, with theoretical guarantees that match the standard threat model in terms of cost and effectiveness.

- We validate our methods on the real-world dataset, demonstrating that even sparse, bounded fake data can significantly bias bandit learners in practice.

Our work bridges the gap between theoretical models of adversarial bandits and practical data-driven attacks observed in real systems. It introduces new threat models, techniques, and insights that we hope will inspire more realistic evaluations of online learning algorithms in adversarial environments.

**Related Work.** There is a growing body of work on adversarial attacks against bandit algorithms Jun et al. (2018); Liu and Shroff (2019); Garcelon et al. (2020); Ma and Zhou (2023); Wang et al. (2022). Jun et al. (2018) showed that with sublinear cost, an attacker can steer UCB and $\epsilon$-greedy toward suboptimal arms. Building on this, Liu and Shroff (2019) extended the paradigm to unknown algorithms and contextual bandits, while Garcelon et al. (2020) studied perturbations to both contexts and rewards in linear bandits. More recently, Zuo (2024) introduced optimal attack strategies against UCB and Thompson Sampling, establishing matching lower bounds. Most of these works assume a strong *feedback-perturbation* model with direct reward modifications each round Jun et al. (2018); Liu and Shroff (2019); Garcelon et al. (2020); Zuo (2024), or bounded variants that still allow continuous interference Xu et al. (2021); Wang et al. (2024); Zuo et al. (2023). Another line of research has focused on defense algorithms against adversarial attacks and corruptions, including robust MAB (Lykouris et al. (2018); Gupta et al. (2019)), robust linear bandits (Bogunovic et al. (2021); He et al. (2022)), and combinatorial bandits (Xu and Li (2021)). However, the corruption-based threat models considered in these works are typically weaker than those studied in the attack literature, since the adversary is not allowed to observe the learner's actions before corrupting the feedback.

We differ from both lines by introducing the *Fake Data Injection Threat Model*, where adversaries cannot tamper with genuine rewards but may inject fabricated (arm, reward) pairs. This setting reflects realistic constraints in recommendation or advertising platforms, where fake users can be created but authentic feedback remains immutable. Beyond bandits, it parallels data poisoning in broader ML systems (recommenders, ads, A/B testing, RL), and our tools (e.g., the Exponential Suppression Lemma) provide general insights into vulnerabilities of sequential decision-making.

## 2 PRELIMINARIES

### 2.1 STOCHASTIC BANDITS

We consider the standard stochastic multi-armed bandit setting with the arm set $[K] := \{1, 2, \ldots, K\}$, where each arm $k \in [K]$ is associated with an unknown reward distribution with mean $\mu_k$. The reward distributions are assumed to be $\sigma^2$-sub-Gaussian, with $\sigma^2$ known. Without loss of generality, we assume the arms are ordered such that $\mu_1 \geq \mu_2 \geq \cdots \geq \mu_K$. In each round $t = 1, 2, \ldots, T$, the learner selects an arm $a_t \in [K]$ and receives a stochastic reward $r_t$ drawn from the distribution corresponding to arm $a_t$. In this paper, we consider two widely used algorithms for stochastic bandits: Upper Confidence Bound (UCB) and Thompson Sampling (TS).

**Upper Confidence Bound (UCB)**. We consider the UCB algorithm as specified in Jun et al. (2018); Zuo (2024), and the prototype is the $(\alpha, \psi)$ algorithm of Bubeck and Cesa-Bianchi (2012, Section 2.2). In the first $K$ rounds, the learner pulls each arm once to initialize reward estimates. For subsequent rounds $t > K$, the learner selects the arm with the highest UCB index $a_t = \arg\max_{i \in [K]}[\hat{\mu}_i(t) + 3\sigma\sqrt{(\log t)/N_i(t)}]$, where $\hat{\mu}_i(t)$ is the empirical mean reward of arm $i$ and $N_i(t)$ is the number of times arm $i$ has been selected up to round $t$.

**Thompson Sampling (TS)**. We consider the Thompson Sampling algorithm specified in Zuo (2024), and the prototype is the $(\alpha, \psi)$ algorithm of Agrawal and Goyal (2017). In the first $K$ rounds, the learner pulls each arm once. For rounds $t > K$, a sample $\nu_a$ is drawn independently for each arm $a \in [K]$ from the distribution $\mathcal{N}(\hat{\mu}_a(t), 1/N_a(t))$, and the learner selects the arm with the largest sampled value $a_t = \arg\max_{a \in [K]} \nu_a$. In the absence of attacks, both UCB and TS are known to achieve sublinear regret by selecting suboptimal arms only $o(T)$ times.

### 2.2 PREVIOUS THREAT MODEL AND LIMITATIONS

We begin by reviewing the standard threat model adopted in prior works on adversarial attacks against bandit algorithms Jun et al. (2018); Liu and Shroff (2019); Zuo et al. (2023); Zuo (2024). In each round $t$, the learner selects an arm $a_t$ to play, and the environment generates a pre-attack reward $r_t^0$ drawn from the underlying distribution of arm $a_t$. The attacker then observes the tuple $(a_t, r_t^0)$ and decides an attack value $\alpha_t$. The learner can only receive the post-attack reward $r_t = r_t^0 - \alpha_t$. Define the cumulative attack cost as $C(T) = \sum_{t=1}^{T} |\alpha_t|$. The attacker's objective is to manipulate the learner into selecting a specific target arm for a linear number of rounds while incurring only sublinear attack cost. Formally, an attack is considered successful if the attacker can force the learner to select the target arm $T - o(T)$ times while ensuring that $C(T) = o(T)$. While many attack strategies have been proposed under the standard threat model, it exhibits several critical limitations when applied to practical settings.

*First*, the model assumes that the attacker can perturb the environment-generated reward in *every* round. This assumption is often unrealistic in real-world applications such as recommender systems. For example, consider an app that recommends restaurants to users and collects their feedback to improve future recommendations. An attacker may wish to bias the system toward recommending a particular restaurant, but it is infeasible to directly modify the feedback of all real users. In practice, a more common attack strategy is to create fake users who submit fabricated feedback. However, even this is constrained by operational or detection limits—adding fake users in every round is highly impractical. Thus, the assumption of per-round attack capability does not reflect realistic adversarial power. *Second*, the model restricts the attacker to modifying only the reward of the chosen arm in each round. In contrast, fake-user-based attacks offer more flexibility. A fake user can submit feedback on any item (i.e., any arm), regardless of what the learner selected in that round. This means fake data injection enables the attacker to fabricate feedback for arbitrary arms, not just the one currently played by the learner. As a result, the standard model underestimates the attacker's flexibility in practice and overestimates their ability to act at every timestep. *Third*, the threat model assumes that both the pre-attack reward and the attack values are unbounded. In many systems, user feedback is naturally bounded — e.g., binary click signals or discrete rating scores (e.g., 1 to 5 stars). Allowing arbitrarily large attack values could result in out-of-range or clearly invalid feedback, which would either be filtered out by the system or easily flagged as suspicious. Therefore, attacks that rely on large reward perturbations are incompatible with these bounded-feedback environments.

## 3  NEW THREAT MODEL: FAKE DATA INJECTION

To address practical limitations of the standard adversarial attack model, we introduce a new and more realistic threat model, which we call the *Fake Data Injection Threat Model*. This model captures how adversaries behave in real-world systems such as recommendation platforms, where direct manipulation of genuine user feedback is infeasible, and attacks are often carried out by injecting fabricated interactions (e.g., fake users with fake feedback).

In the Fake Data Injection model, the attacker does *not* interfere with the feedback received by the learner during normal interactions. Instead, the attacker is allowed to inject up to $N^F$ *fake data samples*, denoted by $\{(a_i^F, r_i^F)\}_{i \in [N^F]}$, into the learner's history. Each fake data point mimics a legitimate user interaction, where $a_i^F \in [K]$ is the selected arm and $r_i^F \in [\tilde{a}, \tilde{b}]$ is the corresponding reward for two given bounds $\tilde{a} \leq \tilde{b}$. Crucially, the learner treats these injected samples as if they were genuine past interactions: it updates the empirical mean $\hat{\mu}_{a_i^F}$, increments the pull count $N_{a_i^F}$, and advances its internal round counter $t$. For example, if two fake samples are injected at round $t$, the learner behaves as though the arms were played at rounds $t$ and $t + 1$, and then resumes its online interaction at round $t + 2$. We define the total attack cost as

$$C^F(T) := \sum_{i \in [N^F]} \left| r_i^F - \mu_{a_i^F} \right|,$$

and consider an attack *successful* if it can mislead the learner into pulling a target arm for $T - o(T)$ rounds while ensuring $C^F(T) = o(T)$.

This hybrid mechanism distinguishes our model from both prior online perturbation models and offline poisoning settings. In offline data poisoning (e.g., Liu and Shroff (2019)), the attacker modifies a fixed dataset collected from arms actually played, and the learner is then trained once on this corrupted dataset. In contrast, our attacker can inject arbitrary $(a, r)$ pairs at any time, affecting arms that may never have been pulled. The learner immediately incorporates these fake samples and continues to make decisions in an online, round-by-round fashion. Thus, while our model permits batch-style updates, it remains fundamentally online, bridging the gap between classical reward-perturbation attacks and purely offline poisoning. It resolves several key limitations of the previous threat model:

**Limited access.** The attacker cannot modify every round's reward but can only add a finite number of fake interactions, as in practice, creating new accounts or reviews incurs cost and detection risk.

**Cross-arm flexibility.** Fake data can target any arm, not just the one chosen by the learner, reflecting real-world scenarios where attackers can inject reviews or clicks on arbitrary items.

**Bounded realism.** Injected rewards must lie in the valid range $[\tilde{a}, \tilde{b}]$, ensuring plausibility such as binary clicks or star ratings(e.g., 1 to 5 stars) and avoiding unrealistic unbounded perturbations.

## 4  ATTACK STRATEGIES

In this section, we develop attack strategies specifically designed for the fake data injection threat model. We begin by studying a simplified setting with unbounded feedback, which serves as a conceptual bridge from prior threat models: instead of directly altering the learner's observed rewards, the attacker injects fake data with unrestricted values. It allows us to highlight the core mechanisms and intuitions behind effective attack strategies. Building on these, we then consider constrained injection attacks, where both the *magnitude* of fake feedback and the *injection frequency* are limited. Our goal is to design strategies that balance these two constraints while still steering the learner toward suboptimal behavior efficiently. Without loss of generality, we assume that the target arm is arm $K$ which has the lowest expected reward.[1]

### 4.1  WARM-UP: INJECTION ATTACKS WITH UNBOUNDED FEEDBACK

We begin our study of fake data injection attacks by considering a relaxed setting in which the injected reward values $r_i^F$ can take *arbitrary real values*, i.e., the bounded feedback constraint from

---

[1]This represents the most challenging case for the attacker and can be easily extended to target any other arm.

Section 3 is removed. This setting closely mirrors the standard threat model, where the attacker can directly modify the observed reward as $r_t = r_t^0 - \alpha_t$, allowing unbounded perturbations $\alpha_t$ in each round. In this unbounded injection setting, we demonstrate that *injecting a single fake data point per non-target arm* is sufficient to mislead the learner into favoring the target arm. We formalize this insight through the Least Injection Algorithm, a simple yet effective one-shot attack strategy against the UCB algorithm, as shown in Algorithm 1.

---

**Algorithm 1:** Least Injection Algorithm on UCB

**Input:** Attack parameter $\delta_0 > 0$

1 **for** *round* $t = 1, 2, \dots$ **do**
2    **for** *each non-target arm* $i \in [K-1]$ **do**
3       **if** *arm* $i$ *has not been attacked* **and** $N_i(t) = \lceil (\log T)/\delta_0^2 \rceil$ **then**
4          Inject fake data sample:
5          $(a_i^F, r_i^F) = \left( i, N_i(t) \cdot \left( \hat{\ell}_K(t) - \hat{\mu}_i(t) \right) + \hat{\ell}_K(t) \right)$
6          $t \leftarrow t + 1$
7    **end**
8    **end**
9 **end**

---

The attack operates as follows. For each non-target arm $i$, we wait until it has been pulled $N_i(t) = \lceil (\log T)/\delta_0^2 \rceil$ times. At this point, we inject a single fake sample designed to reduce its empirical mean below a high-probability lower bound of the target arm. Specifically, we define the empirical lower confidence bound for target arm $K$ as:

$$\hat{\ell}_K(t) := \hat{\mu}_K(t) - 2\beta(N_K(t)) - 3\sigma\delta_0,$$

where $\beta(N) := \sqrt{\frac{1}{2N} \log \frac{\pi^2 K N^2}{3\delta}}$, and $\delta_0 > 0$ is a tunable attack parameter. The injected sample on non-target arm $i$ ensures that after the attack, the empirical mean of arm $i$ satisfies:

$$\hat{\mu}_i(t+1) \leq \hat{\ell}_K(t), \tag{1}$$

thus making it unlikely to be selected in future rounds. The total number of injected fake data points is at most $K-1$, one per non-target arm. Define $\Delta_i := \mu_i - \mu_K$. We now provide the formal theoretical guarantee of Algorithm 1.

**Theorem 4.1.** *Suppose $T > 2K, \delta < 0.5$. With probability at least $1 - \delta$, Algorithm 1 forces the UCB algorithm to select the target arm in at least*

$$T - \mathcal{O}\left( (K-1)(\log T)/\delta_0^2 \right)$$

*rounds, using a cumulative attack cost of at most*

$$C^F(T) = \sum_{i=1}^{K-1} |r_i^F - \mu_i| \leq \mathcal{O}\left( \sum_{i=1}^{K-1} (\Delta_i + 4\beta(1) + 3\sigma\delta_0) \cdot \frac{\log T}{\delta_0^2} \right).$$

Compared with the attack algorithm under the standard threat model in Jun et al. (2018), the Least Injection Algorithm achieves a similar level of target-arm selection with comparable sublinear attack cost. Notably, the parameter $\delta_0$ controls the trade-off between the number of non-target arm pulls and the attack cost: increasing $\delta_0$ reduces the number of non-target pulls but increases the cost per injection. However, the marginal benefit diminishes once $\delta_0 > \sqrt{\log T}$, beyond which the cost grows without improving effectiveness. By selecting $\delta_0 = \Theta(\sqrt{\log T})$, the cumulative attack cost is minimized to $\widehat{\mathcal{O}}(K\sigma\sqrt{\log T})$, which matches the lower bound $\Omega(\sqrt{\log T})$ established in Zuo (2024).

To prove Theorem 4.1, we introduce the following lemma, which plays a central role in our attack design and serves as a key building block for subsequent algorithms.

**Lemma 4.1** (Exponential Suppression of Non-Target Arms). *Suppose $T > 2K, \delta < 0.5$. With probability at least $1 - \delta$, for any non-target arm $i \in [K-1]$ that has been pulled $N_i(t)$ times, if a fake data point is injected according to Line 5 of Algorithm 1, then arm $i$ will not be selected again until at least round $\exp(N_i(t)\delta_0^2)$.*

*Proof Sketch.* After the injection, the empirical mean of arm $i$ is reduced such that its UCB index becomes significantly lower than that of the target arm. We analyze the evolution of the UCB indices and show that, unless arm $i$ is pulled again (which it is not), its confidence bound tightens slowly while its empirical mean remains suppressed. By induction over subsequent rounds, we show that the UCB index of arm $i$ remains lower than that of the target arm for an exponential number of rounds, specifically up to round $\exp(N_i(t)\delta_0^2)$. □

**Remark.** *Lemma 4.1 establishes a critical property of our attack strategy: once a non-target arm $i$ has been pulled sufficiently and a properly chosen fake data point is injected, its UCB index becomes exponentially suppressed. More precisely, if the following two conditions are satisfied (1) $N_i(t) \geq (\log T)/\delta_0^2$ and (2) $\hat{\mu}_i(t+1) \leq \hat{\ell}_K(t)$, then arm $i$ will not be selected again until after round $T$. This suppression effect is crucial: it guarantees that once the attack is applied to arm $i$, its influence on the learning process becomes negligible for the remaining rounds. The attacker can thus prevent further exploration of non-target arms using only a single injection per arm, ensuring that the learner increasingly concentrates on the target arm. This mechanism forms the backbone of all our attack strategies.*

In addition to attacking UCB, we extend the Least Injection Algorithm to target the Thompson Sampling algorithm. Specifically, the attacker injects a single fake data point into each non-target arm $i$ when $N_i(t) = \lceil (\log T)/\delta_0^2 \rceil$, using a modified version of Line 5 in Algorithm 1. Due to space limitations, we defer the full algorithm and details to the appendix. We provide its theoretical guarantee below.

**Theorem 4.2.** *Suppose $T > 2K$, $\delta < 0.5$. With probability at least $1 - 2\delta$, the modified Least Injection Algorithm forces the Thompson Sampling algorithm to select the target arm in at least $T - \mathcal{O}((K-1)\log T/\delta_0^2)$ rounds, using a cumulative attack cost of at most $\mathcal{O}\big(\sum_{i=1}^{K}(\Delta_i + 4\beta(1) + \sqrt{8\log\left((\pi^2 K)/(3\delta)\right)} + 4\sqrt{\log T})(\log T)/\delta_0^2\big)$.*

Compared with the attack algorithm under the standard threat model studied in Zuo (2024), our approach achieves a similar level of target-arm selection with matching attack cost. By setting the attack parameter $\delta_0 = \Theta(\sqrt{\log T})$, we obtain a total cost of $\mathcal{O}(\sqrt{\log T})$, which aligns with the known lower bound, demonstrating the near-optimality of our strategy under this relaxed fake data injection model.

### 4.2 Constrained Injection Attacks

We now turn to more realistic and constrained settings where injected fake data must lie within a bounded range. This reflects practical scenarios in which user feedback—such as clicks or ratings—is inherently limited (e.g., binary or on a fixed scale). To address the constraint on individual fake rewards, we first propose a natural extension of Algorithm 1. In this version, the influence of a single unbounded fake reward is approximated by injecting *a batch of bounded fake samples* simultaneously for each non-target arm.

In practice, however, attackers may face an additional constraint on the number of fake samples that can be injected at any given time, due to resource limitations or system-level detection thresholds. To address this more challenging setting, we introduce a periodic injection strategy that operates under two constraints: (1) each fake reward must be bounded; (2) only a limited number of fake samples can be injected at once. Our strategy carefully coordinates the frequency and timing of injections to maintain effective adversarial influence over the learner while satisfying both constraints. Despite these limitations, we show that the attacker can still successfully manipulate the learner's behavior with sublinear cost.

### 4.2.1 Simultaneous Bounded Injection

To ensure the attack remains realistic and stealthy, we consider a practical setting in which each injected fake reward must lie within a bounded range $[\tilde{a}, \tilde{b}]$. Under this constraint, we propose the Simultaneous Bounded Injection (SBI) algorithm, which extends Algorithm 1 by replicating the effect of a single unbounded fake reward via the injection of multiple bounded fake samples. Specifically, for each non-target arm, the attacker injects a batch of rewards with the minimum value $\tilde{a}$ in a single round to achieve the same suppression of the empirical mean as in the unbounded setting.

---

**Algorithm 2:** Simultaneous Bounded Injection on UCB

---

**Input:** Attack parameter $\delta_0$, bounded reward range $[\tilde{a}, \tilde{b}]$

1 **for** *round $t = 1, 2, \ldots$* **do**
2      **for** *each non-target arm $i \in [K-1]$* **do**
3          **if** *arm $i$ has not been attacked* **and** $N_i(t) = \lceil (\log T)/\delta_0^2 \rceil$ **then**
4              $\tilde{n} \leftarrow \left\lceil \frac{\hat{\mu}_i(t) - \hat{\ell}_K(t)}{\hat{\ell}_K(t) - \tilde{a}} \cdot \left\lceil \frac{\log T}{\delta_0^2} \right\rceil \right\rceil$;
5              Inject $\tilde{n}$ fake samples $(i, \tilde{a})$;
6              $t \leftarrow t + \tilde{n}$
7          **end**
8      **end**
9 **end**

---

As shown in Algorithm 2, once a non-target arm $i$ has been pulled $\lceil (\log T)/\delta_0^2 \rceil$ times, the attacker computes $\tilde{n}$ and injects $\tilde{n}$ fake samples with reward $\tilde{a}$. We make the following assumption to ensure that the suppression of arm $i$'s empirical mean is always feasible:

**Assumption 4.1.** $\tilde{a} \leq \mu_K - 3\beta(1) - 3\sigma\delta_0$.

This assumption ensures suppression remains feasible under bounded fake rewards. Prior works (e.g., Jun et al. (2018); Liu and Shroff (2019); Zuo (2024)) assume unbounded manipulation, where feedback can be reduced arbitrarily and a margin condition holds implicitly. In practice, the requirement can be relaxed to $\tilde{a} < \hat{\mu}_K(t) - 2\beta(N_K(t)) - 3\sigma\delta_0$ at a specific round $t$. We regard this as a mild assumption, since it only requires the target arm's mean to exceed the minimum possible reward by a small margin. We now present the theoretical guarantee of the SBI algorithm.

**Theorem 4.3.** *Suppose $T > 2K, \delta < 0.5$ and Assumption 4.1 hold. With probability at least $1 - \delta$, Algorithm 2 forces the UCB algorithm to select the target arm in at least*

$$T - \mathcal{O}\left( \sum_{i=1}^{K-1} \frac{\mu_i + \beta\left((\log T)/\delta_0^2\right) - \tilde{a}}{\mu_K - 3\beta(1) - 3\sigma\delta_0 - \tilde{a}} \cdot \frac{\log T}{\delta_0^2} \right)$$

*rounds, using a cumulative attack cost of at most*

$$\mathcal{O}\left( \sum_{i=1}^{K-1} (\mu_i - \tilde{a}) \frac{\Delta_i + 4\beta(1) + 3\sigma\delta_0}{\mu_K - 3\beta(1) - 3\sigma\delta_0 - \tilde{a}} \cdot \frac{\log T}{\delta_0^2} \right).$$

Compared with Theorem 4.1, the attacker now injects $\tilde{n}$ fake samples for each non-target arm instead of a single one. While this increases the total number of injected samples, the order of the attack cost remains $\mathcal{O}(\sqrt{\log T})$ when the attack parameter is set to $\delta_0 = \Theta(\sqrt{\log T})$. Thus, the SBI algorithm maintains asymptotic optimality under more realistic constraints.

As in Section 4.1, the SBI algorithm can also be extended to attack the Thompson Sampling algorithm. Due to space constraints, we defer the full algorithm and details to the appendix and present the main theorem below.

**Theorem 4.4.** *Suppose $T > 2K, \delta < 0.5$. With probability at least $1 - 2\delta$, the modified Simultaneous Bounded Injection forces the Thompson sampling algorithm to select the target arm in at least*

$$T - \mathcal{O}\left( \sum_{i=0}^{K-1} \frac{\mu_K - 3\beta(1) - \sqrt{8\log(\pi^2 K/(3\delta))} - 4\sqrt{\log T} - \tilde{a}}{\mu_i + \beta\left((\log T)/\delta_0^2\right) - \tilde{a}} \cdot \frac{\log T}{\delta_0^2} \right) \tag{2}$$

*rounds, using a cumulative attack cost of at most*

$$\mathcal{O}\left( \sum_{i=1}^{K-1} (\mu_i - \tilde{a}) \frac{\Delta_i + 4\beta(1) + \sqrt{8\log(\pi^2 K/(3\delta))} + 4\sqrt{\log T}}{\mu_K - 3\beta(1) - \sqrt{8\log(\pi^2 K/(3\delta))} - 4\sqrt{\log T} - \tilde{a}} \cdot \frac{\log T}{\delta_0^2} \right). \tag{3}$$

### 4.2.2 PERIODIC BOUNDED INJECTION

The SBI algorithm above assumes that the attacker can inject all required fake samples within a single round. However, this assumption may not hold in practice. For example, in a restaurant

recommendation system, injecting a large batch of fake (e.g., low-rating) reviews at once may trigger anomaly detection mechanisms, leading the system to filter or ignore the fake data. In contrast, injecting smaller amounts of fake feedback periodically—at a controlled rate—can be significantly less suspicious and more effective in practice.

To model this scenario, we introduce a more restrictive and realistic setting where:

1. The attacker can inject at most $f$ fake samples in any single round (batch size constraint);

2. Consecutive injections on the same arm $i$ must be separated by a cooldown period of at least $R_i$ rounds, where $R_i$ depends on the maximum batch size $f$.

To address this setting, we propose the Periodic Bounded Injection (PBI) algorithm, shown in Algorithm 3. Given a maximum batch size $f$, the algorithm adaptively schedules periodic injections to suppress the empirical mean of non-target arms while respecting both constraints.

---

**Algorithm 3:** Periodic Bounded Injection on UCB

**Input:** Attack parameter $\delta_0$, reward bound $[\tilde{a}, \tilde{b}]$, max batch size $f$

1 **for** *round* $t = 1, 2, \ldots$ **do**
2    **for** *each non-target arm* $i \in [K-1]$ **do**
3      **if** *arm $i$ has not been attacked* **and** $N_i(t) = \lceil (\log T)/\delta_0^2 \rceil$ **then**
4        $\tilde{n}_i \leftarrow \left\lceil \frac{\hat{\mu}_i(t) - \hat{\ell}_K(t)}{\hat{\ell}_K(t) - \tilde{a}} \cdot \left\lceil \frac{\log T}{\delta_0^2} \right\rceil \right\rceil$;
5        $R_i \leftarrow \min_{1 \le c \le \lceil \frac{\tilde{n}_i}{f} \rceil} \frac{1}{c} \exp\left( \left( \left( \frac{\hat{\mu}_K(t) - 2\beta(N_K(t)) - \tilde{\mu}_i(t_i(c))}{3\sigma} \right)^2 \cdot (N_i(t) + fc) \right) \right) - t - f$;
6        $\text{next}_i \leftarrow t$;
7      **end**
8      **if** $\tilde{n}_i > 0$ **and** $\text{next}_i \le t$ **then**
9        Inject $f$ fake samples $(i, \tilde{a})$;
10        $\tilde{n}_i \leftarrow \tilde{n}_i - f$;
11        $t \leftarrow t + f$;
12        $\text{next}_i \leftarrow \text{next}_i + f + R_i$;
13      **end**
14    **end**
15 **end**

---

The PBI algorithm distributes the injection of fake samples across multiple rounds rather than injecting them all at once. Once a non-target arm $i$ reaches the designated pull threshold ($\lceil (\log T)/\delta_0^2 \rceil$), the attacker computes both the total number of fake samples $\tilde{n}_i$ required to suppress the empirical mean of arm $i$, and a waiting interval $R_i$, which ensures that the fake samples can be injected periodically without allowing arm $i$ to regain a high UCB index. The notation $\tilde{\mu}_i(t + f)$ represents the estimated value of $\hat{\mu}_i(t + f)$ with $f$ fake data injections starting from round $t$. At each interval of $R_i + f$ rounds, a batch of $f$ fake samples is injected until the total $\tilde{n}_i$ is exhausted. This strategy effectively balances *stealthiness* and *attack efficacy*, making it robust against detection in practical systems with bounded feedback and rate-limited injection constraints.

The analysis of cumulative attack cost for PBI is deferred to the Appendix, as it is similar to that of Theorem 4.3: the total number of fake samples injected remains the same. What distinguishes PBI is how *suppression is maintained across time*, which is guaranteed by the following lemma.

**Lemma 4.2.** *The choice of $R_i$ in Algorithm 3 ensures that once a batch of $f$ fake data samples is injected into non-target arm $i$, the arm will not be selected again for at least the next $R_i$ rounds.*

*Proof Sketch.* This result builds on a modified version of the exponential suppression lemma (Lemma 4.1). Rather than suppressing a non-target arm with a single large injection, we analyze the suppression effect of a partial injection of $f$ bounded fake samples. We show that the first batch induces the weakest suppression, so it suffices to compute $R_i$ based on this worst-case scenario. By ensuring that the UCB index remains below that of the target arm during this interval, we guarantee that arm $i$ is not selected within the next $R_i$ rounds. After the $c$-th period of injection,

we have $\hat{\mu}_i \le \hat{\mu}_K - 2\beta(N_K(t)) - 3\sigma\sqrt{(\log(t+(f+R)c))/N_i(t+(f+R)c)}$. And its UCB index will remain lower than arm $K$'s until at least round $t+(f+R)c$. $\qquad\square$

We also extended the PBI algorithm to attack the Thompson Sampling algorithm. Due to space limitations, we defer the detailed algorithm and corresponding results to the appendix.

# 5 EXPERIMENTS

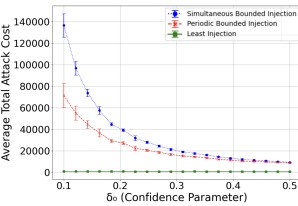 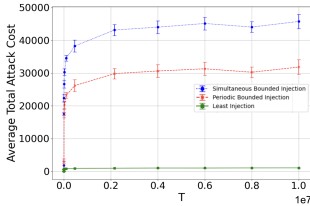 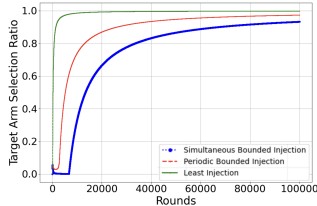

Figure 1: Attack Costs vs $\delta_0$      Figure 2: Attack Costs vs $T$      Figure 3: Selection Ratios

We evaluate our attack strategies in a realistic setting using the MovieLens 25M dataset Harper and Konstan (2015), which reflect the practical motivations of the Fake Data Injection model. Due to space constraints, we report results for SBI, PBI, & LI algorithms on the UCB learner; other setting results are in the appendix. We consider $K = 10$ arms and simulate user interaction traces with stochastic rewards derived from movie rating distributions. The time horizon is set to $T = 100,000$. For PBI, the per-round injection limit is set to $f = 5$. Figure 1 plots the average attack cost as a function of $\delta_0$, the confidence parameter. As expected, increasing $\delta_0$ reduces the number of required fake samples for suppressing non-target arms, leading to lower attack costs. PBI consistently incurs lower cost than SBI for small $\delta_0$ due to its more conservative, distributed injection schedule. Figure 2 shows the total attack cost over the time horizon $T$. Across all settings, PBI performs comparably or better than SBI, while Least Injection achieves the lowest overall cost due to unbounded injection values. SBI and PBI are highly effective in Figure 3 settings, with the learner converging to the target arm in most rounds after early exploration. The PBI strategy achieves particularly strong empirical performance while incurring lower attack cost than SBI. While both SBI and PBI still achieve similar levels of effectiveness as the LI baseline under realistic constraints, the results highlight the practical advantage of temporally distributed attacks.

# 6 CONCLUDING REMARKS

This work introduces a practical and realistic threat model, Fake Data Injection, for adversarial attacks on stochastic bandits. In contrast to prior models that assume per-round, unbounded reward perturbations, our framework captures real-world constraints such as bounded feedback, limited injection capability, and the attacker's inability to modify genuine user data. Within this model, we develop a suite of effective attack strategies that successfully manipulate both UCB and TS algorithms using only sublinear-cost injections. Our theoretical analysis and experimental results demonstrate that even sparse, bounded fake interactions can significantly bias stochastic bandit algorithms.

Despite these results, several limitations remain and open avenues for future work. We assume a passive learner that processes all feedback without defense, which may not hold in robust or adversarial-aware systems. Our work also focuses on stochastic bandits; extending to contextual or reinforcement learning settings remains an open challenge. Additionally, real-world attackers may face detection risks or adaptive filtering by the system—scenarios not captured in our current framework. Future work should explore defense mechanisms such as anomaly detection or arm-level auditing, and investigate the dynamic interplay between attackers and adaptive learners. Addressing these limitations will be crucial for building secure online learning systems in adversarial environments. Due to space limitations, we defer the detailed discussion of potential defenses to the Appendix.

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
