# OpenReview forum: "Practical Adversarial Attacks on Stochastic Bandits via Fake Data Injection"
_ICLR.cc/2026/Conference — Submitted to ICLR 2026_

### Official Review · Reviewer_GGiN · 2025-10-30

**Soundness:** 2
**Presentation:** 3
**Contribution:** 2
**Rating:** 2
**Confidence:** 3

**Summary:**

This paper proposes a new attack model in MAB called Fake Data Injection (FDI). Unlike the standard threat model, in FDI, the attacker can not arbitrarily manipulate the feedback, and cannot attack in every round, which is closer to the real world scenario. Attack strategies are given for MAB and TS algorithm, and there are theoretical bounds of the trade-off between attack cost and attack performance.

While the motivations make sense to me, I don't feel that all of them are precisely reflected through the proposed attack model, and I'm not able to fully differentiate it from the existing attack model. Another limitation is that the attack is specific and customized to MAB and TS algorithms. For details please refer to ``weakness''.

**Strengths:**

1. The attack model takes more constraints and is closer to real-world application.

**Weaknesses:**

1. My main concern is the applicability of this model, because it needs to keep track of number of pulls. I don't think this can be known to the attacker in the real world?
2. Attack is limited to and customized for MAB and TS. In practice, it's infeasible to know exactly the recommendation algorithm. Can we extend this to, at least a class of algorithms?
3. Around Line 179, seems to me that the success of attack is still defined in terms of the amout of total corruption. In other words, whether $N^F$ is large or not seems not to matter, which seems inconsistent with real-world motivation. In this case, how is this different from the standard successful attack definition in the literature?
4. Is it possible to examine the performance of some existing corruption-robust algorithms? If not, what is the challenge? (This is just for curiosity, and I do NOT ask for this during the rebuttal)

Would like to increase the rating if these are addressed by the author(s)

**Questions:**

1. In Sec. 4.2.2, how would $f$ and $R_i$ affect the tradeoff in Thm. 4.4?

---

> ### Author Response · Authors · 2025-11-21
>
> > **W1. Applicability of the model: keeping track of the number of pulls.**
>
> **A1.**
> We thank the reviewer for raising this question. We would like to clarify that assuming the attacker can track the number of pulls per arm is standard across the attack literature (Jun et al. 2018; Liu \& Shroff 2019; Garcelon et al. 2020; Ma \& Zhou 2023; Wang et al. 2022). All of these works require the attacker to observe aggregate statistics, such as empirical means or pull counts, to plan targeted manipulations. Our model adopts exactly the same information structure and is fully aligned with this established line of work.
>
> In practice, such statistics are often directly available or easily inferred: rating platforms (IMDb, Yelp, Amazon), app stores (Google Play, Apple App Store), and recommendation systems routinely display counts such as number of ratings, number of reviews, or number of clicks. These publicly visible statistics correspond exactly to the pull counts and empirical means in the bandit formulation. Indeed, real attackers such as fake-review farms and click-injection services use these publicly available counts to decide when and how to inject influence.
>
> Thus, the assumption that the attacker can track the number of pulls is both consistent with prior attack papers and realistically satisfied in common online platforms. Our model follows this widely adopted framework and remains applicable in realistic attack scenarios.
>
> > **W2. Extension to general bandit algorithms.**
>
> **A2.**
> We thank the reviewer for highlighting this important direction. Following prior work such as Jun et al. (2018) and Zuo (2024), our focus in this paper is to analyze the vulnerability of representative and widely used bandit algorithms, UCB and TS, under our proposed practical threat model. We agree that extending the analysis to general bandit algorithms is a valuable next step. Liu et al. (2019, Section 4.2) propose a general attack strategy that targets arbitrary bandit algorithms. However, their approach assumes that the attacker can act in every round and manipulate the observed reward of the chosen arm, which is not compatible with our more realistic fake data injection model.
>
> Nevertheless, we believe that our core technique, especially the exponential suppression effect in Lemma 4.1, can potentially be combined with their strategy. In particular, their result (Theorem 4) relies on the assumption that any suboptimal arm is pulled at most $O(\log T)$ times without attack. In our case, if the attacker can obtain a tighter (possibly instance-dependent) upper bound $N_i(T)$ on suboptimal arm pulls, then a suppression-based strategy could be designed to ensure that each non-target arm is eliminated after $N_i(T)$ rounds by injecting appropriately timed fake data. While the precise design of such a general attack remains an open problem, we believe our suppression mechanism provides a promising foundation toward this goal. We appreciate the reviewer's suggestion and will consider it as a future research direction.
>
>
> > **W3. Definition of attack success and the role of $N^F$.**
>
> **A3.**
> We thank the reviewer for the helpful question. We clarify that $N^F$ is not a pre-given budget or a success criterion. In our fake-data-injection model, $N^F$ is simply the total number of fake samples produced by the algorithm. Because fake rewards cannot be arbitrarily small, the attacker may need multiple injections to achieve the same empirical-mean shift that a classical attack could obtain with a single unbounded perturbation. Thus, $N^F$ is the number of injections necessitated by these bounded constraints, rather than an externally tunable parameter.
>
> The definition of attack success is unchanged from the standard literature: the target arm should be played for $T-o(T)$ rounds with controlled total attack effort. What differs is how the effort is quantified. In the classical model, effort comes solely from the cumulative magnitude of per-round reward perturbations. In our setting, effort is carried by both the number of injections and their (bounded) magnitudes, because fake samples affect the empirical mean only through incremental contributions. Viewed this way, the injection-based cost is directly analogous to the classical corruption cost, but adapted to a more realistic manipulation channel. If the attacker instead has a fixed injection budget, our suppression lemma characterizes exactly when that budget suffices to eliminate a non-target arm. We will clarify these points in the revision.

---

> ### Author Response · Authors · 2025-11-21
>
> > **W4. Is it possible to examine the performance of some existing corruption-robust algorithms?**
>
> **A4.**
> We thank the reviewer for the question. Unfortunately, the robust MAB algorithms proposed in Lykouris et al. (2018) and Gupta et al. (2019) are not effective under our adversarial injection threat model.
>
> Their setting, often referred to as the corruption model, assumes the adversary only has access to the learner’s history up to the previous round, but not the learner’s action in the current round. To exploit this limitation, their algorithms introduce randomness in action selection as a defense mechanism. In contrast, our model, like prior attack-based threat models (e.g., Jun et al. 2018), assumes a stronger adversary who knows not only the full history but can also control both the action and reward of injected samples. In our setting, the attacker independently decides which fake data to inject, making randomized defenses (like those in Lykouris et al. and Gupta et al.) ineffective. For empirical evidence, we refer to Figure 4 in Wang et al. (2025), which shows that robust algorithms such as MLAAER and BARBAR fail under the standard attack model.
>
> (Wang et al. 2025) X. Wang, M. Liu , J. Zuo , X. Liu , J. C.S. Lui, and M. Hajiesmaili. Stochastic Bandits Robust to Adversarial Attacks. In ICLR, 2025.
>
> > **Q1. How would $f, R_i$ affect the tradeoff in the Theorem?**
>
> **A5.**
> We thank the reviewer for the question. We clarify that neither the batch size $f$ nor the cooldown parameter $R_i$ affects the total number of fake samples needed to suppress arm $i$; this quantity $n_i$ is determined solely by the empirical-mean gap that must be closed (and by standard concentration parameters) and is therefore independent of these scheduling parameters. In our setting, $f$ is an input that specifies how many fake samples may be injected at a time, and our analysis shows how to schedule the required $n_i$ injections under any such batching constraint. Thus, $f$ influences only the temporal pattern of injections, i.e., how the $n_i$ samples are partitioned over time, while the overall injection cost remains unchanged.

---

### Official Review · Reviewer_kj36 · 2025-11-02

**Soundness:** 4
**Presentation:** 4
**Contribution:** 4
**Rating:** 6
**Confidence:** 5

**Summary:**

This paper studied fake data injection attacks on UCB and Thompson sampling algorithms. Different from prior works that allow attacker to perturb the reward in every step, this paper considered a more practical attack scenario where the attacker can only inject fake data in certain steps to guarantee sparseness and boundedness. The paper starts a motivating attack that injects a single but very negative reward to reduce empirical mean of non-target arms, and then extended this attack by distributing this big attack to multiple rounds and periodic stages. The paper provided theoretical analysis to show that even under sparse and bounded attack scenario, the attacker is still able to force the bandit to always pull some target arm while incurring only sublinear attack cost.

**Strengths:**

The paper extended traditional bandit attacks to a more realistic setting, where the attacker can only inject fake data in limited number of steps. This kind of attack is more stealthy and less likely to be detected.

The paper derived solid theoretical results to show the effectiveness and efficiency of the proposed attack algorithms.

The paper performed empirical study of the attacks on real-world data to show that the performance of the attack.

Overall, the paper is sound, solid, and provided a comprehensive investigation of the problem.

**Weaknesses:**

Fake data injection, although more stealthy, may make the data suspicious due to the injection. Traditional attacks does not alter the actions of the bandit learner, but only changes the reward. However, fake data injection causes a change in the action selected by the bandit algorithm. For example, if a batch of fake data all have the same action and are injected, then the learner may suddenly receive a set of data with exactly the same action, but the learner only chooses to take that action once. This data definitely result in an alarm and the learner may realize that it is being attacked. From this perspective, the fake data injection might as well be more suspicous.

**Questions:**

What if the bandit algorithm finds fake data injection suspicious? How to address it and how to justify fake data injection attack is more realistic and stealthy?

---

> ### Author Response · Authors · 2025-11-21
>
> > **W1 \& Q1. What if the bandit algorithm finds fake data injection suspicious? How to address it and how to justify fake data injection attack is more realistic and stealthy?**
>
> **A1.**
> We thank the reviewer for raising this concern. We agree that naively injecting a large batch of identical samples could indeed look suspicious. This is precisely why our three attack schemes (Single Injection → SBI → PBI) are designed to progressively reduce detectability. Single Injection performs direct injection, SBI spreads the injections over time, and PBI further limits injections to a small fraction of rounds. Furthermore, the injected samples do not need to be identical: the attacker may draw them from a small interval around the intended value so that each point individually appears plausible, while their average achieves the desired shift in empirical means. Thus the model naturally accommodates randomized, low-volume, and staggered injections that avoid the “sudden batch” patterns highlighted by the reviewer.
>
> The fake data injection framework is motivated by realistic applications where feedback comes from many independent users, e.g., rating platforms, recommendation logs, mobile-app interactions. In such systems, attacks typically occur through multiple accounts or sessions, each contributing only a few benign-looking interactions. Modern anomaly detection focuses on outliers within *each* account (e.g., high posting frequency, repeated identical actions), whereas small, randomized contributions distributed across many accounts generally evade such defenses. Our model captures this aggregated, cross-user manipulation, and our SBI/PBI algorithms explicitly incorporate mechanisms (temporal spreading, limited-round injection, randomized values) to operate within realistic stealth constraints. We will clarify these points and add examples to make the practical relevance of fake data injection more explicit.
>
> Looking forward, we see several natural extensions: integrating explicit anomaly-detection models into the attack design; studying attacker–defender dynamics where the learner adapts to suspicious patterns; and analyzing the trade-off between attack effectiveness and detectability under platform-specific constraints. These directions would further strengthen the realism and applicability of fake data injection models, and we view our current formulation as a clean foundation for these future studies.

---

### Official Review · Reviewer_eRxK · 2025-11-03

**Soundness:** 1
**Presentation:** 2
**Contribution:** 1
**Rating:** 2
**Confidence:** 5

**Summary:**

This work studies adversarial attacks on stochastic multi armed bandits (MABs).
In prior works, the adversary was only allowed to perturb the rewards obtained by the algorithm for the arm is actually picked during the online process.
In this work, the authors introduce a new notion for adversarial attacks on MABs called Fake Data Injection where the adversary is allowed to inject any arm and reward pair to MAB algorithm and algorithm considers it as if it chose the arm and updates it's beliefs.
The authors claim that this attack model is more realistic as it can allow the adversary to attack only in limited rounds and also attack any arbitrary arm instead of just the arm chosen by the model. This could simulate situations such as fake reviews or fake engagement for any specific arm/content without the algorithm even recommending that arm in the currrent round. The authors also consider that the adversary cannot manipulate in every round but only a small fractions of rounds.

The authors consider attacks on UCB and Thompson sampling.

First, the authors consider an unbounded reward model and show that if the adversary can add unbounded corruptions, then it can make the algorithm pick the target arm in all but $o(T)$ rounds using only $\tilde{O}({K \sigma \sqrt{\log T}})$ noise.

Then the authors consider the case of bounded corruptions, and show that under certain assumptions on the bounds of the corruption, the algorithm from unbounded setting can be updated to make multiple bounded attacks every time it needs to attack to get similar bounds on the corruption to ensure the model selects the target arm in all but $o(T)$ rounds.

The authors then consider a followup where the attacker can only inject atmost $f$ attacks in one go show how the attack can be adopted for the setting.

The authors also provide experiments where they simulate the setup on MovieLens25 dataset and show that their attack methods do work for UCB and with limited attack cost, the algorithm can indeed be tricked into selecting the target arm for most rounds.

**Strengths:**

- The paper introduces and interesting model for data corruption attacks where the adversary can attack any arm
- The consider the setting where the attacker has constraints on the bound of the rewards
- They have experiments using MovieLens 25M showing the effectiveness of the algorithm

**Weaknesses:**

- My main concern for this paper is that I believe the main results are not true and there is a bug in the analysis.

In the proof for Theorem 4.1. Let's consider any specific arm $i$, since each arm is corrupted once, the cost of corrupting $i$ according the definition of the corruption is $| r_i - \mu_i|$ where $r_i$ is the corrupted reward.

To ensure that after the corruption, eq (1) holds, we need
\begin{align*}
    & \hat{\mu}(t+1) \leq \hat{l}_k(t) \\
    \implies & \frac{\hat{\mu}_i(t) N_i(T) + r_i}{N_i(T) + 1} \leq  \hat{l}_k(t) \\
     \implies & r_i \leq \hat{l}_k(t) (N_i(T) + 1) - \hat{\mu}_i(t) N_i(T)
\end{align*}

Now consider corruption $| r_i - \mu_i|$. Since we are attacking the arm, it's obvious that we are trying to reduce it's mean, so $r_i - \mu_i$ is infact negative and $|r_i - \mu_i| = \mu_i - r_i$ not $r_i - \mu_i$

So we have
\begin{equation}
     r_i \leq \hat{l}_k(t) (N_i(T) + 1) - \hat{\mu}_i(t) N_i(T)
\end{equation}
and
\begin{equation}
    C =  \mu_i - r_i
\end{equation}
Combining the two we get
\begin{equation}
    C \geq \mu_i + \hat{\mu}_i(t) N_i(T) - \hat{l}_k(t) (N_i(T) + 1)
\end{equation}
This is completely opposite of what is claimed in the proof for Theorem 4.1 in Appendix Section A.2.2

- I don't agree with the claim that this model has more restrictions as it allows corrupting only a fraction of rounds not all. In the algorithm design, all the algorithms proposed by the authors are also just arbitrarily corrupting whenever they want. In fact I believe this problem is simpler as the learner does not have to analyze whether the arm would be picked by the algorithm or not and can inject noise whenever and wherever.

- I don't agree with the claim that earlier works only considered the unbounded reward setting. Xu et all 2021 consider rewards bounded between 0 and 1. Moreover in the bounded case, the authors assume an upper bound on the lower limit of the boundary which is not well motivated.

**Questions:**

Can the proofs be fixed?
Can you formally compare the difficulty of this attack model with the existing model studied where we are only allowed to corrupt the arms picked by the algorithm?

---

> ### Author Response · Authors · 2025-11-21
>
> > **W1. Correctness of the Proof for Theorem 4.1.**
>
> **A1.**
> We appreciate the reviewer’s detailed calculation and effort in checking our analysis. The concern arises from a *logical inversion* in how the inequality
> $$
> \hat{\mu}_i(t+1) \le \hat{\ell}_K(t)
> $$
>
> is used. In Algorithm 1, the attacker does *not* determine the injected reward by manipulating this inequality. Instead, the algorithm *defines* the injected reward so that the inequality holds with *equality*, namely
> $$
> (a_i^F, r_i^F)=\bigl(i,\; N_i(t)\bigl(\hat{\ell}_K(t)-\hat{\mu}_i(t)\bigr)+\hat{\ell}_K(t)\bigr),
> $$
>
> which ensures by construction that
> $$
> \hat{\mu}_i(t+1)=\hat{\ell}_K(t).
> $$
>
> The reviewer’s lower bound derivation is mathematically correct for any $r_i$ satisfying the inequality, but it does not apply to our analysis because we do not optimize over this inequality; we simply choose the unique $r_i$ that forces equality.
>
> Our attack-cost analysis (Lines 704--719) then upper-bounds
> $$
> C_i^F(T) = |r_i^F - \mu_i|
> $$
>
> and sums over all non-target arms, using standard high-probability concentration of $\hat{\mu}_i$ and $\hat{\mu}_K$, exactly as in prior attack works (e.g., Jun et al. 2018; Zuo 2024). This approach follows the classical attack strategy of resetting each non-target arm’s empirical mean to the target arm’s confidence level, rather than pushing it lower.
>
> In short, we do not derive our upper bound from the inequality $\hat{\mu}_i(t+1)\le \hat{\ell}_K(t)$; we first choose the injected value that enforces equality, and then analyze the resulting attack cost separately. The reviewer’s argument analyzes a different mechanism and therefore does not contradict our results. We would be happy to clarify further if the reviewer has additional questions.
>
> > **W2 \& Q1. The fake data injection attacker can add fake samples whenever and to whichever arm, regardless of the learner’s action. How does this compare to existing attack models?**
>
> **A2.**
> We thank the reviewer for the thoughtful question. Our intention is not to claim that the fake data injection model is strictly “more restrictive” or “harder” than the classical reward‐perturbation attack model. Rather, the two models offer different adversarial capabilities, and neither is a strict subset of the other. In the classical model, the attacker can freely overwrite every observed reward, exercising full per-round control. In contrast, fake data injection cannot alter real feedback at all and must rely on a small number of bounded synthetic samples whose influence must persist through many uncontaminated updates. The challenge is therefore not predicting when an arm will be pulled, but determining how to time and size a few injections so that their effect lasts, despite long intervals without intervention. This motivates our progression from Single Injection → SBI → PBI, each accommodating increasingly realistic constraints such as bounded fake rewards and per-round limits.
>
> For comparability with existing attacks, our analysis yields explicit expressions for the injection required to suppress non-target arms. These expressions carry over directly to the classical setting: when a non-target arm is pulled, inserting the same amount of synthetic feedback (or equivalently modifying the reward) produces the same suppression effect, so our suppression lemma and cost bounds extend with only minor adjustments.
>
> In summary, the fake data injection model is not meant to replace or dominate classical attacks, but to provide a more realistic and stealth-aware threat model with different constraints, and our contribution is in showing how to operate effectively under these practical limitations. We are happy to clarify further if needed.

---

> ### Author Response · Authors · 2025-11-21
>
> > **W3. Earlier works studied bounded rewards.**
>
> **A3.**
> We thank the reviewer for the comment. We would like to clarify that we do *not* claim prior work only considered unbounded rewards. As stated in our Related Work, we explicitly cite bounded-reward variants (Xu et al. 2021; Wang et al. 2024; Zuo et al. 2023).
>
> However, even under bounded rewards, the threat models in these papers differ fundamentally from ours. For example, in Xu et al. (2021), the adversary may still overwrite the entire reward vector on any attacked round with arbitrary values in $[0,1]^K$, enabling direct per-round interference with the learner’s updates. Likewise, Zuo et al. (2023) continues to rely on per-round modification and provides only asymptotic, gap-dependent cost bounds, whereas our analysis yields finite-time guarantees. These settings therefore capture a form of continuous reward manipulation that is not present in our fake data injection model.
>
> In contrast, as discussed in A2, our adversary cannot alter environment-generated feedback at all and is restricted to inserting a small number of bounded synthetic samples that must compete with long stretches of uncontaminated observations. The key technical difficulty lies in ensuring that these sparse injections have lasting influence, a challenge addressed by the suppression lemma and the progression from Single Injection to SBI and PBI.
>
> In short, while bounded rewards have appeared before, bounded rewards combined with limited-frequency, history-only injections define a different and more practical threat model, and our contribution is in analyzing how to conduct effective attacks under these realistic constraints.
>
> > **W4. Assumption on the upper bound of the lower reward limit.**
>
> **A4.**
> The assumption on the upper bound of the lower reward limit in our bounded-injection setting (Assumption 4.1) is required only to ensure that suppression is feasible. In the bounded case, the attacker cannot push rewards arbitrarily low; therefore, to guarantee that a non-target arm’s empirical mean can be driven below the decision boundary, we need a small margin such as
> $$
> \tilde{a} \le \mu_K - 3\beta(1) - 3\sigma\delta_0,
> $$
> which ensures that every non-target arm can eventually be suppressed relative to the target arm. This condition is mild: in bounded-reward bandits it is common to assume that the best arm’s mean is not exactly at the minimum reward value, so that it can be distinguished from the others.
>
> We also note that prior attack works (Jun et al. 2018; Liu and Shroff 2019; Zuo 2024) implicitly avoid this issue by assuming unbounded or arbitrarily low perturbations, so the required margin always exists by construction. In our bounded-injection model, where rewards cannot be reduced arbitrarily, this explicit margin is the natural analogue of that implicit assumption.

---

### Official Review · Reviewer_ZQ1n · 2025-11-08

**Soundness:** 1
**Presentation:** 3
**Contribution:** 1
**Rating:** 2
**Confidence:** 4

**Summary:**

This paper studied the adversarial attacks on stochastic bandits. It designed and analyzed attack strategies which aim to prevent superior performance of classical TS and UCB algorithms and presented some numerical results.

**Strengths:**

The paper begined with an introduction, discussed some related works, proposed attack strategies which are accopanied by theoretical results and experimental results.

**Weaknesses:**

My main concern is about the motivation and hence contribution of this work. The author(s) **grounded the work on the statement that the existing works are with unrealistic assumptions on adversarial attacks**. The details are listed in the second paragraph in Section 2.2 and my corresponding concerns are as below:
1. 'Attacker can perturb the environment-generated reward in every round.' --- It is incorrect. Some existing works, such as \url{https://link.springer.com/article/10.1007/s10994-018-5758-5}, assume that the attacker can only attack a fraction of time steps, instead of all time steps.
2. 'The attacker modifies the reward of only the selected arm' --- It is incorrect for some existing works. Some papers such as \url{https://proceedings.mlr.press/v99/gupta19a/gupta19a.pdf} assume that the agent pull and arm only after attacks, in which case it is impossible for the attacker to only attack the selected arm.
3. 'The pre-attack reward and the attack values are unbounded' --- It is incorrect for some existing works. Bounded-reward case is usually a much easier one and there has been many works on that. Again, check the results in \url{https://proceedings.mlr.press/v99/gupta19a/gupta19a.pdf}

Based on my concerns above,
1. I am a bit confused why this fake-data-injection setting is interesting to explore and what is its difference from the existing setting. It seems to me that the amount of injection is similar to the amount of corruption.
2. Moreover, the numerical/analytical performance of proposed strategies should be compared to existing ones, which I failed to find in the manuscript.
Overall, I suggest the author(s) to discuss more related works and convince the readers the necessity of this formulation and contribution of this work.

**Questions:**

In addition to my concerns raised in the **Weaknesses* section, I suggest the author(s) to be careful about citation formatting. For instance, in line 30, there should be '(Li et al., 2010)' instead of 'Li et al. (2010)'.

---

> ### Author Response · Authors · 2025-11-21
>
> > **W1. Unrealistic assumptions in existing adversarial attack works.**
>
> **A1.**
> We appreciate the reviewer’s detailed comments. The three concerns you raised (perturbing rewards every round, modifying only the selected arm, and unbounded rewards) arise from conflating two distinct research directions in stochastic bandits: the adversarial attack literature and the corruption-robust (defense) literature. As stated in our Related Work: prior adversarial attack papers “assume a strong feedback-perturbation model with direct reward modifications each round  (Jun et al. 2018; Liu and Shroff 2019; Garcelon et al. 2020; Zuo 2024), or bounded variants that still allow continuous interference (Xu et al., 2021; Wang et al., 2024; Zuo et al., 2023)”, whereas “the corruption-based threat models considered in these works are typically weaker than those studied in the attack literature, since the adversary is not allowed to observe the learner’s actions before corrupting the feedback”. The papers cited in the comments (Kapoor et al. 2019; Gupta et al. 2019) belong to the latter defense line rather than the attack line that our model explicitly extends.
>
> As emphasized in Wang et al. (2025) and Zuo (2024), corruption-robust models fundamentally cannot capture targeted manipulation under a strong, adaptive adversary; in fact, no algorithm can be robust against such an adversary with sufficient attack budget (Fact 1 in Zuo (2024); also see A4 to Reviewer GGIN). Hence, although the defense setting assumes a weaker adversary and correspondingly imposes fewer restrictions on the corruption model, the results from that line cannot be transferred to the attack setting. Our contribution lies within the attack literature: we point out that existing attack models rely on unrealistic assumptions (per-round reward rewriting, modification limited to the selected arm, and unbounded perturbations), and we propose Fake Data Injection as a more realistic mechanism that relaxes these assumptions while preserving the essential adversarial adaptivity. For this reason, corruption-robust models are not directly relevant to the threat model we critique or extend.
>
> (Wang et al. 2025) X. Wang, M. Liu , J. Zuo , X. Liu , J. C.S. Lui, and M. Hajiesmaili. Stochastic Bandits Robust to Adversarial Attacks. In ICLR, 2025.
>
> > **W2. The amount of injection is similar to the amount of corruption.**
>
> **A2.**
> We thank the reviewer for the thoughtful question. As clarified in our response to W1, the corruption-robust (defense) setting is fundamentally different from the adversarial attack setting, so here we focus on the distinction between fake data injection and the classical reward-perturbation threat model used in prior attack works.
>
> Although our total attack cost matches the asymptotic order in Zuo (2024), the underlying problem in our setting is substantially more constrained. In the classical attack model, the adversary can overwrite the observed reward in every round and can apply arbitrarily large perturbations; designing an optimal attack reduces to choosing the appropriate per-round corruption. In contrast, fake data injection limits the adversary to a finite number of bounded synthetic samples, which cannot be injected every round and must be timed strategically. Achieving comparable attack cost under these strict constraints requires progressively addressing more difficult scenarios. We first show that in the unbounded case, a single injection per non-target arm can replicate the effect of classical attacks. Under bounded rewards, the SBI algorithm approximates this effect through carefully sized batches. Finally, the PBI algorithm handles the most realistic setting with bounded feedback and per-round injection limits, requiring periodic scheduling to maintain long-term suppression. None of these injection-scheduling or magnitude-constraint challenges arise in prior attack models, which is exactly why fake data injection is compelling: it captures realistic limitations while still enabling near-optimal adversarial influence.

---

> > ### Comment · Reviewer_ZQ1n · 2025-11-23
> >
> > Thanks for your response. However, my fundamental concerns are not addressed.
> >
> > The three things I mentioned (perturbing rewards every round, modifying only the selected arm, and unbounded rewards) are what were maintained by the author(s) as **unrealistic assumptions in existing adversarial attack works**.
> >
> > However, I dont see that existing works are with all these unrealistic assumptions and hence I am still a bit confused why this fake-data-injection setting is interesting to explore and what is its difference from the existing setting.

---

> ### Author Response · Authors · 2025-11-21
>
> > **W3. Numerical/analytical comparison with existing methods.**
>
> **A3.**
> We thank the reviewer for the suggestion. As noted in our response to Reviewer eRxK (A4), our fake data injection framework can be naturally extended downward to recover existing formulations, and our algorithms remain valid in those settings. However, the reverse is not true: existing attack methods cannot be used in our Fake Data Injection model, because their threat model relies on capabilities that are explicitly unavailable here. Classical attacks assume per-round reward manipulation with unrestricted magnitude, and their mechanisms require repeatedly overwriting the learner’s observed reward. Under our constraints, finite injections, bounded samples, and per-round injection limits, these methods simply cannot execute the operations they rely on, and therefore cannot serve as meaningful baselines in our experiments.
>
> For analytical comparison, we note that the order of attack cost we obtain matches the optimal bound established in prior attack models (e.g., Zuo 2024), despite operating under a significantly more restricted threat model. This highlights that our design is competitive even though the attacker is much weaker. We will revise the manuscript to provide a clearer comparison to relevant related work and articulate how our results relate to, and differ from, the classical attack literature.
>
> > **Q1. Citation formatting issue.**
>
> **A4.**
> We thank the reviewer for pointing this out. We have corrected the citation formatting in the revised version.

---

> > ### Comment · Reviewer_ZQ1n · 2025-11-23
> >
> > Thanks for your response.

---

> ### Author Response · Authors · 2025-11-24
>
> We thank the reviewer for the follow-up comment. We would like to explicitly clarify that **all** prior works in the adversarial attack line (not the corruption-robust/defense line), including Jun et al. (2018), Liu \& Shroff (2019), Garcelon et al. (2020), Zuo (2024), Xu et al. (2021), Wang et al. (2024), and Zuo et al. (2023), operate under the **direct reward–perturbation** model and inherently rely on the first two assumptions (perturbing rewards every round; modifying only the selected arm). While some of these works partially studied bounded rewards (e.g., Xu et al. 2021; Wang et al. 2024; Zuo et al. 2023; see also A3 to eRxK for more detailed discussion), the underlying mechanism remains unchanged: the attacker sits between the environment and the learner and directly rewrites the observed feedback. As discussed in Section 2.2, we view this direct reward–manipulation model as unrealistic in scenarios where feedback comes from many independent users (e.g., recommendation platforms), where no adversary can feasibly intercept and alter every real user interaction.
>
> By contrast, our new threat model removes this privileged control position entirely. In fake data injection, the attacker never rewrites environment-generated rewards; instead, it behaves like an ordinary user contributing additional data to the system’s historical record. This modeling shift fundamentally changes the nature of the attack problem: without direct overwrite power, the adversary must induce influence through limited numbers of bounded samples whose effects are diluted by large volumes of clean observations. This makes the **allocation and timing** of fake samples central to the problem, which motivates our progression from Single Injection to SBI to PBI, each step accommodating increasingly realistic constraints such as bounded sample values and per-round injection limits. Since classical attack strategies rely on direct manipulation of reward feedback, they cannot be used in this injection setting. Our model therefore enables a different class of adversarial strategies, better aligned with realistic manipulations such as fake reviews, injected clicks, and fabricated user interactions. In this work, we study these constraints jointly through this progressive framework toward increasingly realistic and low-visibility attack behaviours.
>
> In summary, prior attack works assume direct, privileged control of the reward stream, whereas our model assumes no such access and requires influencing the learner indirectly through limited, naturalistic insertions. If any part of this distinction remains unclear, or if further elaboration would be helpful on any specific point, we would be very happy to provide additional clarification.

---

### Author Response · Authors · 2025-12-03

We thank all reviewers for their thoughtful feedback and the Area Chair for handling our submission. The primary concerns focused on our comparison to existing work (Reviewer ZQ1n), the validity of our analysis (Reviewer eRxK), and the practical applicability of our threat model (Reviewers kj36, GGiN). In response, we have provided detailed, point-by-point answers to each comment, and we believe our clarifications thoroughly address these issues.

> **Comparison with existing adversarial attack literature.** (Reviewers ZQ1n, eRxK)

There are two main lines of work in this area: (1) the corruption-robust direction mentioned by Reviewer ZQ1n (e.g., Kapoor et al. 2019; Gupta et al. 2019) and (2) the adversarial attack direction (e.g., Jun et al. 2018, Liu \& Shroff 2019, Garcelon et al. 2020, Zuo 2024). The former assumes a weaker, history-only adversary and focuses on designing defenses against such corruption, whereas the latter studies a stronger, adaptive adversary that is standard in adversarial-attack settings. Our paper fits into this adversarial-attack category: we clarify that all prior works in this line operate under the direct reward-perturbation model and therefore implicitly rely on unrealistic assumptions (e.g., being able to perturb rewards every round and modify only the selected arm). We argue that these assumptions do not reflect real-world platforms, where adversaries cannot overwrite genuine user feedback. This motivates our fake-data-injection model, where the attacker is outside the feedback channel and cannot alter environment-generated rewards. This distinction forms the foundation of our contribution.

> **Correctness of our analysis for Theorem 4.1.** (Reviewer eRxK)

The reviewer suggested a bug in our analysis. After a careful re-examination, we found that the concern stems from a logical inversion in how the inequality $\hat{\mu}_i(t+1) \le \hat{\ell}_K(t)$ is interpreted. Our method uses this inequality only to *define* the injected reward; once this value is determined, we analyze the resulting attack cost separately. This analysis follows standard high-probability concentration arguments used in prior works (e.g., Jun et al. 2018; Zuo 2024). Thus, there is no flaw in our reasoning, and the theoretical results of Theorem 4.1 remain valid.

> **Practical applicability of our model.** (Reviewers kj36, GGiN)

We clarify the availability of aggregate statistics such as pull counts or empirical means is standard in the attack literature (e.g., Jun et al. 2018; Liu and Shroff 2019; Garcelon et al. 2020; Zuo 2024) and also realistic in practice: such information is public on almost all rating and recommendation platforms (e.g., number of ratings, number of clicks). Moreover, our model captures realistic constraints absent in prior attack works: (1) bounded rewards, (2) limited-frequency injection and (3) no ability to overwrite observations. Under these constraints, the timing and allocation of fake samples become central challenges. This motivates our methodological progression from Single Injection to SBI to PBI, each step handling increasingly realistic constraints.

**In summary**,

- eRxK, kj36, and GGiN acknowledged that our proposed adversarial attack model is novel and better aligned with real-world applications. While ZQ1n and GGiN sought clarification regarding its relationship to prior attack works, we have now explicitly positioned our contribution within the context of the adversarial attack literature.

- Reviewers ZQ1n, kj36, and GGiN, especially kj36, recognized the strength of our theoretical results. We have carefully re-examined the issue raised by Reviewer eRxK and confirm that there is no flaw in our analysis.

- All reviewers are satisfied with our experimental results.

We believe our work offers the most realistic adversarial model for bandit attacks to date, with potential to inspire future advances in both theoretical developments and practical deployments in this area. Although the initial reviewer scores were diverse, we are confident that our responses have addressed the reviewers’ concerns substantively and constructively. We would greatly appreciate the AC’s reconsideration of the evaluation.

---

### Meta-Review · Area_Chair_Zoh1 · 2026-01-07

**Summary:**

This paper studied adversarial attacks on stochastic multi-armed bandits (MAB). The authors considered a new threat model named Fake Data Injection, which assumes the attacker can only inject a limited number of bounded fake samples into the history while not being able to poison or overwrite other feedback. The authors propose attack strategies (SBI, PBI) and theoretically analyze the success of attack and cost upper bounds against UCB and Thompson Sampling.

While one reviewer (kj36) was positive about the theoretical results, other reviewers raised several concerns, including 1) unfair/incorrect claims regarding the difference in settings/threat models compared to existing literature, which leads to questions about the novelty and significance; 2) practical applicability of the fake data injection; and 3) a concern over the mathematical correctness of Theorem 4.1, which AC believes is correct with authors' clarification in rebuttal. Overall, AC recognized authors' explanation but believes additional clarification and revision is needed to fully address reviewers' concerns.




Reviewer Concerns

**Reviewer Concerns:**

Addressed:

Correctness of Theorem 4.1: Reviewer eRxK raised a concern over the mathematical correctness of Theorem 4.1, which AC believes is correct with authors' clarification in rebuttal.

Practical applicability of the fake data injection: Reviewer GGiN and kj36 shared the concern, which is partially addressed by authors' clarification

Outstanding:

Novelty and signification about setting and techniques: This is a shared concern (ZQ1n, eRxK, GGiN) that the authors made unfair/incorrect claims regarding the difference in settings/threat models compared to existing literature, which leads to questions about the novelty and significance. While the authors provided a distinction between injection and poisoning, reviewer still feels the technical solution was very similar to prior work and technical novelty needs to be further clarified.

**Reviewer Scores:**

Reviewer ZQ1n (2 -> 2): The reviewer remained unconvinced after the first response, remaining concern about the difference with previous attack models.

Reviewer eRxK (2 -> 4): AC believes the reviewer will raise score with authors' clarification over the mathematical correctness of Theorem 4.1, but remain negative due to concern about comparison to existing methods.

Reviewer kj36 (6 -> 6): Remain positive.

Reviewer GGiN (2 -> 4): reviewer may raise score over the explanation of practical applicability of the fake data injection

---

### Decision · Program_Chairs · 2026-01-26

Reject